# GFNet: Homography Estimation via Grid Flow Regression

## Abstract

Current deep homography estimation methods are constrained to processing image pairs with limited resolution due to restrictions in network architecture and computational capacity. For larger images, downsampling is often necessary, which can significantly degrade estimation accuracy. To address this limitation, we propose GFNet, a Grid Flow regression Network that consistently delivers high-accuracy homography estimates across varying image resolutions. Unlike previous methods that directly regress the parameters of the global homography between two views, GFNet directly estimates flow over a coarse grid and then uses the resulting correspondences to compute the homography. This approach not only supports high-resolution processing but also preserves the high accuracy of dense matching while significantly reducing the computational load typically associated with such frameworks, thanks to the use of coarse grid flow. We demonstrate the effectiveness of GFNet on a wide range of experiments on multiple datasets, including the common scene MSCOCO, multimodal datasets VIS-IR and GoogleMap, and the dynamic scene VIRAT. In specific, on GoogleMap, GFNet achieves an improvement of +9.9% in auc@3 while reducing MACs by ∼47% compared to the SOTA dense matching method. Additionally, it shows a 1.7× improvement in auc@3 over the SOTA deep homography method.

## 1 Introduction

Homography estimation is the task of determining the transformation that aligns two planes. It typically finds application in the alignment of two images, each depicting the same planar structure but from different vantage points. This is a fundamental low-level computer vision task widely used in various downstream applications, including image/video stitching (Zhou et al., 2024), multimodal image fusion (Xu et al., 2023), GPS-denied UAV localization (Wang et al., 2024a), stereo vision (Kumar et al., 2024), and planar object tracking (Liu et al., 2023).

Recently, learning-based approaches have demonstrated high accuracy in homography estimation at small image resolutions. The primary reasons for these resolution limitations arise from constraints in (1) *network architecture* and (2) *computational capacity*. Some methods (DeTone et al., 2016; Le et al., 2020; Shao et al., 2021; Cao et al., 2022; 2023; Zhu et al., 2024) compute matching information between image pairs and aggregate it to regress the homography. The aggregation layer, typically using a fixed number of large-kernel pooling operations, outputs a tensor of shape $B \times 2 \times 2 \times 2$ to represent the eight parameters of the homography. This implies that the network's downsampling factor is predefined and constrained by the large kernel sizes, inherently restricting the input size. Other methods (Chang et al., 2017; Zhao et al., 2021; Zhang et al., 2023; Zhang & Ma, 2024) explicitly use the IC-LK iterator (Baker & Matthews, 2004) with learned features. This approach requires multiple iterations in feature space, resulting in significant compute and memory requirements. As a result, current methods are forced to work with inputs at low resolution. Unfortunately, doing so has also a direct negative impact on the accuracy of the estimated homography.

Compared to homography estimation methods, general matchers trained for two- or multi-view correspondence can handle a wider range of image resolutions. However, they tend to underperform in accurately estimating homography. For example, estimating homography between images from different modalities is a common challenge in homography estimation. A case in point is aligning infrared and visible images to aid UAV navigation at night under poor illumination (Li et al.,

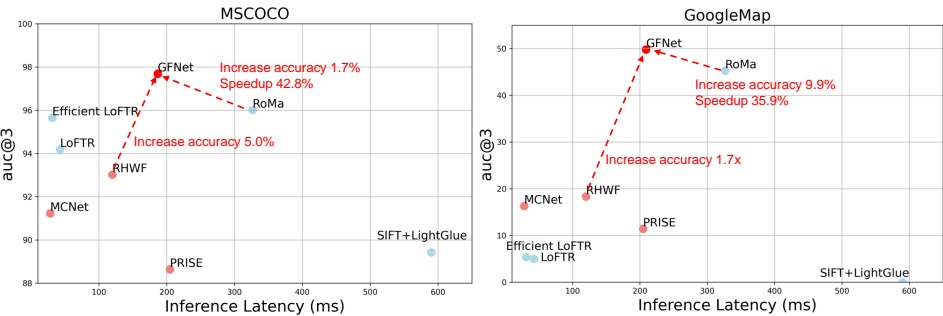

Figure 1: Homography estimation performance of GFNet and existing homography estimation methods (red spot) and image matching methods (blue spot) on MSCOCO (natural) and GoogleMap (multimodal). We test on an RTX3090 with batchsize = 1, resolution = $448 \times 448$.

2023). In such scenarios, sparse (DeTone et al., 2018; Sarlin et al., 2020; Lindenberger et al., 2023; Potje et al., 2024) and semi-dense Sun et al. (2021); Chen et al. (2022); Wang et al. (2024b) matchers often fail to find sufficient good matches in textureless regions (especially in infrared images), finally resulting in poor homography estimation. To address this issue, dense matchers incur in non-negligible computational costs. For example, training the SOTA dense matcher RoMa (Edstedt et al., 2024) on a 24GB GPU only supports a batch size of 1 for $560 \times 560$ images. This heavy resource demand arises from resolving every possible match between images, which is redundant for planar transformations where the displacement field follows a globally smooth pattern.

Currently, there is no homography estimation method that can handle varying image resolution while delivering accurate results without significant computational costs. To address this, we leverage the advantages of resolution flexibility and high accuracy inherent in the dense matching framework, while tackling its computational challenges by considering the globally smooth nature of homography transformations. This leads to our proposed Grid Flow regression Network (GFNet). As shown in Figure 1, GFNet achieves superior accuracy over current homography estimation methods, while outperforming the SOTA dense matching method RoMa with higher efficiency. This is achieved by the following contributions:

- We incorporate pre-trained self-supervised learning features of DINOv2 (Oquab et al., 2023) with a lightweight feature pyramid network to construct robust multi-scale features. This fusion addresses the low-resolution limitation of DINOv2, allowing our model to benefit from its strong cross-domain feature matching capabilities across multiple scales.

- We sparsify the pixel-level dense matching approach to a coarse grid prediction, significantly reducing computational costs while keeping the high accuracy of dense matching.

- We introduce an iterative flow regression approach to prevent suboptimal multi-scale flow optimization in challenging scenes. Besides, a resolution recurrence strategy is used during inference to improve accuracy and enable GFNet to adapt to varying image resolutions.

## 2 RELATED WORKS

**Deep homography estimation**. One of the earliest approaches uses a stacked image pair as input to a VGG-style network to extract matching information between images, followed by cascade average pooling to regress the homography (DeTone et al., 2016). Subsequent methods improve accuracy by cascading multiple VGG-style networks, where the predicted homography from each network is used to warp the images before feeding them into the next network, allowing for progressively more accurate predictions (Erlik Nowruzi et al., 2017; Le et al., 2020). However, using separate networks for refinement often results in suboptimal performance. To address this, Localtrans (Shao et al., 2021) builds multi-scale features within one network and predicts homography in a coarse-to-fine manner. RHWF (Cao et al., 2023) runs with one network recurrently to improve accuracy. There are some approaches improving accuracy by iteratively optimizing the homography, either by explicitly incorporating the Lucas-Kanade algorithm to achieve feature-metric alignment (Chang et al., 2017; Zhao et al., 2021; Zhang et al., 2023; Zhang & Ma, 2024), or by training the network end-to-end, enabling it to learn the optimization process implicitly (Cao et al., 2022; Zhu et al., 2024).

Figure 2: The overview of GFNet. Using multi-scale features, GFNet regresses the grid flow and confidence map progressively from the coarsest scale to the finest.

**General matching methods**. Current approaches can be categorized into sparse, semi-dense, and dense methods. Sparse methods rely on keypoint detection and description, followed by matching the descriptors (DeTone et al., 2018; Sarlin et al., 2020; Deng & Ma, 2022; Lindenberger et al., 2023; Potje et al., 2024). While efficient, they often struggle with accuracy in textureless regions, which are common in homography estimation tasks (Zhang & Ma, 2024). Semi-dense methods degrade less in such areas by bypassing keypoint detection. Instead, they perform global matching at a coarse level, where initial coarse matches that pass the mutual nearest neighbor test are then refined at a finer level. This means if key matches are missed in the coarse stage, they cannot be recovered. Regarding the dense methods, they predict matches for every pixel using flow regression. Although they achieve impressive accuracy, they require substantial computational resources in terms of both time and memory, which limits their applicability in resource-constrained scenarios (Truong et al., 2020; 2023; Edstedt et al., 2023; 2024).

**Common challenges in homography estimation.** The challenges are twofold and arise from practical application requirements, according to recent literature (Nguyen et al., 2018; Le et al., 2020; Zhang et al., 2020; Shao et al., 2021; Zhao et al., 2021; Cao et al., 2022; Zhang & Ma, 2024). The first one is photometric inconsistency caused by changes in illumination or modality, such as images taken at different times, with different sensors, or in different types (Jiang et al., 2021). The second challenge comes from the violation of the homography assumption due to dynamic occlusions.

## 3 METHOD

Our goal is to predict an accurate homography transformation $\mathbf{H}$ that spatially aligns a source image $I_S$ with a target image $I_T$. We achieve this using the proposed GFNet, as shown in Figure 2. Rather than directly regressing the global homography, GFNet follows the dense matching paradigm but estimates the flow over a regular grid and utilizes the resulting correspondences to compute the homography. This process begins by extracting multi-scale features from both images, which are then used to predict grid flow across multiple scales. Next, we introduce the technique in detail.

### 3.1 MULTI-SCALE FEATURE EXTRACTION

DINOv2 has demonstrated exceptional zero-shot cross-domain feature matching capabilities across a variety of vision tasks. Given that our task involves multimodal image pairs, we incorporate the pre-trained DINOv2 into our feature encoder to enhance the robustness of multimodal feature representation. Specifically, DINOv2 divides the input image into $14 \times 14$ patches and generates a feature vector for each patch. As these patch descriptors are typically high-dimensional, we apply a linear projection layer to reduce their dimensionality, setting it to $64$ in our implementation. To further enrich the features with cross-view information, which is crucial for matching, we add a normalized 2D positional encoding (Chen et al., 2022; Cao et al., 2024) and pass the features through stacked cross-attention layers. The resulting feature is denoted as $\mathcal{F}^1 \in \mathbb{R}^{64 \times \lfloor \frac{H}{14} \rfloor \times \lfloor \frac{W}{14} \rfloor}$, where $H$ and $W$ are the height and width of the original image, $\lfloor \cdot \rfloor$ is the round down operation.

Table 1: Complexity analysis of different regression strategies. Time complexity and memory complexity are calculated for computing the global correlation in each scale.

| | Flow shape | | | | | Time complexity | Memory complexity |
|---|---|---|---|---|---|---|---|
| | $l=1$ | $l=2$ | $l=3$ | $l=4$ | $l=5$ | | |
| pixel-based | $\frac{H}{14} \times \frac{W}{14}$ | $\frac{H}{8} \times \frac{W}{8}$ | $\frac{H}{4} \times \frac{W}{4}$ | $\frac{H}{2} \times \frac{W}{2}$ | $\frac{H}{1} \times \frac{W}{1}$ | $O(H^2W^2)$ | $O\left((\frac{1}{14^4} + \frac{1}{8^4} + \frac{1}{4^4} + \frac{1}{2^4} + 1)H^2W^2\right)$ |
| grid-based | $\frac{H}{14} \times \frac{W}{14}$ | $\frac{H}{14} \times \frac{W}{14}$ | $\frac{H}{7} \times \frac{W}{7}$ | $\frac{2H}{7} \times \frac{2W}{7}$ | $\frac{4H}{7} \times \frac{4W}{7}$ | $O\left((\frac{4}{7})^4H^2W^2\right)$ | $O\left((\frac{1}{14^4} + \frac{1}{14^4} + \frac{1}{7^4} + \frac{2^4}{7^4} + \frac{4^4}{7^4})H^2W^2\right)$ |
| grid-based+iteration | $\frac{H}{14} \times \frac{W}{14}$ | $\frac{H}{14} \times \frac{W}{14}$ | $\frac{H}{7} \times \frac{W}{7}$ | $\frac{2H}{7} \times \frac{2W}{7}$ | $\frac{4H}{7} \times \frac{4W}{7}$ | $O\left(N(\frac{4}{7})^4H^2W^2\right)$ | $O\left(N(\frac{1}{14^4} + \frac{1}{14^4} + \frac{1}{7^4} + \frac{2^4}{7^4} + \frac{4^4}{7^4})H^2W^2\right)$ |

Since DINOv2 produces relatively low resolution features, relying solely on these features for homography estimation would lead to limited matching accuracy (Barroso-Laguna et al., 2024). To address this issue, we introduce a Feature Pyramid Network (FPN) to construct multi-scale features, ranging from the original image size down to a $1/8$ scale. We keep the network lightweight by configuring the channel numbers of each layer to $[8, 16, 32, 64]$. Furthermore, to improve the representation capability of the lightweight FPN, we bilinearly upsample $\mathcal{F}^1$ to the $1/8$ scale and merge it with the $1/8$ scale features produced by the encoding stage of the FPN. The final output features used for homography estimation consist of five scales, denoted as $\{\mathcal{F}^l\}_{l=1,2,3,4,5}$, corresponding to spatial sizes of $1/14$, $1/8$, $1/4$, $1/2$, and $1$ relative to the original image, as illustrated in Figure 2.

## 3.2 BACKGROUND: DENSE FLOW REGRESSION

Before introducing our grid flow regression method, we briefly explain the dense matching framework we follow Truong et al. (2023); Edstedt et al. (2023; 2024). Dense matching aims to find pixel-wise correspondences between two images $I_S, I_T \in \mathbb{R}^{H \times W \times 3}$ by estimating a dense displacement field, or flow, $\mathbf{w} \in \mathbb{R}^{H \times W \times 2}$. Given the pixel coordinates in $I_S$ as $\mathbf{x} \in \mathbb{R}^{H \times W \times 2}$, the relationship $\mathbf{y} = \mathbf{x} + \mathbf{w}$ denotes the corresponding pixel positions in $I_T$ that align with those in $I_S$, representing the same physical locations in the scene. Typically, $\mathbf{w}$ is predicted in a multi-scale manner, from the coarsest ($l = 1$) to the finest:

$$\mathbf{w}^l = \text{up}(\mathbf{w}^{l-1}) + \Delta\tilde{\mathbf{w}}^l, \quad \Delta\tilde{\mathbf{w}}^l = \text{decoder}_\theta^l\left(\mathcal{F}_S^l, \mathcal{F}_T^l, \text{up}(\mathbf{w}^{l-1})\right), \tag{1}$$

where $\theta$ represents learnable parameters, $\mathbf{w}^0$ is an all-zero field, and $\text{up}(\cdot)$ is the bilinear upsampling operation. There is a feature correlation layer included in each $\text{decoder}_\theta^l(\cdot)$, which is computed as:

$$c(\mathcal{F}_S^l, \mathcal{F}_T^l; r^l) = \mathcal{F}_S^l[\mathbf{x}^l] \odot \mathcal{F}_T^l[\mathbf{x}^l + \text{up}(\mathbf{w}^{l-1}) + \delta], \tag{2}$$

where $\mathbf{x}^l \in \mathbb{R}^{H^l \times W^l \times 2}$ is the pixel coordinates in $\mathcal{F}_S^l$, $[\cdot]$ represents bilinear interpolation, $\odot$ represents the dot product, and $\delta \in [-2r^l - 1, 2r^l + 1] \times [-2r^l - 1, 2r^l + 1]$ defines a local window with radius $r^l$. This operation generates a 4D tensor of shape $\mathbb{R}^{H^l \times W^l \times (2r^l+1) \times (2r^l+1)}$, which captures visual similarity within a local neighborhood. While this correlation step is crucial for producing accurate results, it also introduces the main computational bottleneck in dense flow regression.

## 3.3 GRID FLOW REGRESSION

**Grid-based strategy for efficiency**. To alleviate the computational burden of Eq. 2, previous works typically reduce the radius $r$ as the scale increases (Truong et al., 2020; Shao et al., 2021; Zhu et al., 2024). However, this saving is limited for high-resolution images, where the first two dimensions of $c(\mathcal{F}_S^l, \mathcal{F}_T^l; r^l)$ dominate the computational load. So to further minimize computational costs, we leverage the global smooth pattern of homography transformations and focus on learning a sparser representation of $\mathbf{w}$ to replace the original dense, pixel-wise one.

To achieve this, we sparsify $\mathbf{w}$ through a grid-based strategy. We redefine $\mathbf{x} \in G \times G \times 2$ to represent the coordinates of a regular grid on $I_S$ with $G \times G$ being the grid size. Our goal is to estimate a grid-based displacement field $\mathbf{w} \in G \times G \times 2$ and a confidence map $\mathbf{m} \in G \times G \times 1$ that reflects the reliability of each flow estimate. The initial grid size is determined by the resolution of the coarsest feature map, with the grid size scaling proportionally to the feature map scale gap. In our case, we regard the results produced at the coarsest scale, where DINOv2 is applied, as an initialization step. Therefore, we keep the grid size constant for the first two scales and increase it only in the subsequent scales. We list the flow shape and computational complexity of pixel-based and grid-based strategies in Table 1. Theoretically, utilizing a grid-based strategy can markedly reduce the computational burden.

**Direct regression vs. Iterative regression**. The decoder($\mathcal{F}_S^l, \mathcal{F}_T^l, \text{up}(\mathbf{w}^{l-1})$), which estimates the flow only once at each scale (as shown in Eq. 1), is referred to as direct regression. This is the default regression paradigm in dense matching. However, as the convergence basin shrinks with increasing scale, obtaining an optimal estimation with this approach requires that the estimate from the previous level falls within the convergence basin of the next level. If this condition is not met, the result will be suboptimal, as illustrated in Figure 3. The precondition for obtaining a reliable estimate at each scale is that $\mathcal{F}_S^l$ and $\mathcal{F}_T^l$ must be well-aligned with $\text{up}(\mathbf{w}^{l-1})$. This alignment is relatively easy to achieve for images within the similar modality. But

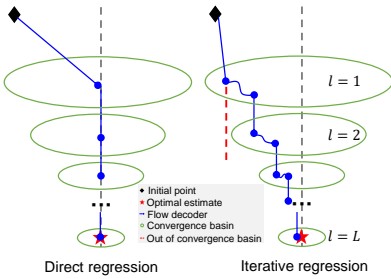

Figure 3: Flow optimization across multiple scales.

in cases where images have significant representational gaps, it becomes challenging for the decoder to make an accurate estimate in a single pass. To address this issue, we introduce an iterative regression strategy to improve the reliability of each grid flow. Specifically, our formulation for estimating the grid flow and its confidence map is given by:

$$\mathbf{w}^{l,n} = \mathbf{w}^{l,n-1} + \Delta\tilde{\mathbf{w}}^{l,n}, \quad \mathbf{m}^{l,n} = \mathbf{m}^{l,n-1} + \Delta\tilde{\mathbf{m}}^{l,n}, \quad \Delta\tilde{\mathbf{w}}^{l,n}, \Delta\tilde{\mathbf{m}}^{l,n} = \text{decoder}_\theta^l\left(\mathcal{F}_S^l, \mathcal{F}_T^l, \mathbf{w}^{l,n-1}\right), \quad (3)$$

where $n \in [1, N]$ represents the iteration number, with $N$ total iterations per scale. When $N = 1$, Eq. 3 reduces to direct regression. All predictions are made at the image scale, and the flow and confidence map are updated between scales as $\mathbf{w}^{l+1,0}, \mathbf{m}^{l+1,0} = \text{up}(\mathbf{w}^{l,N}), \text{up}(\mathbf{m}^{l,N})$. As shown in Table 1, the increase in time and memory complexity due to iterations is negligible when $N$ is small. Ablation studies will demonstrate the significant advantages of the iterative strategy.

**Adaptive resolution recurrence**. During training, images are consistently resized to a fixed resolution before being fed into the network to maintain a uniform grid shape. To handle varying resolutions, we adopt a simple recurrent strategy similar to Truong et al. (2020) during inference:

$$\mathbf{w}_1, \mathbf{m}_1 = \text{GFNet}(I_S, I_T; \mathbf{w}_0), \quad \mathbf{w}_0, \mathbf{m}_0 = \text{GFNet}(I_S', I_T'; \mathbf{0}), \quad I_S', I_T' = \text{resize}(I_S, I_T), \quad (4)$$

where $I_S$ and $I_T$ are the original images, $I_S'$ and $I_T'$ are resized to match the training resolution, and GFNet($\cdot; \mathbf{w}$) denotes GFNet is initialized with $\mathbf{w}$. The linear combination of $\mathbf{m}_0$ and $\mathbf{m}_1$ is regraded as the final confidence map. For images with resolutions lower than the training resolution, we use the training resolution recurrently, while for higher-resolution images, we apply the recurrence at a higher resolution. Different from Truong et al. (2020), our recurrence strategy operates on a grid level rather than at the pixel level, which significantly reduces computational overhead. Additionally, while computing $\mathbf{w}_0, \mathbf{m}_0$ spans all scales, the process of computing $\mathbf{w}_1, \mathbf{m}_1$ begins at the $1/8$ scale, bypassing the $1/14$ scale. This is because the previous stage's $\mathbf{w}_0, \mathbf{m}_0$ provides sufficient initialization, making re-initialization at the $1/14$ scale unnecessary.

In addition to employing the recurrence strategy to improve accuracy during inference, we also adopt a symmetric approach. This involves swapping the input sequence of $I_S$ and $I_T$ to compute the flow for the regular grid on $I_T$. The resulting flows, $\mathbf{w}_{S\to T}$ and $\mathbf{w}_{T\to S}$, along with their corresponding confidence maps $\mathbf{m}_{S\to T}$ and $\mathbf{m}_{T\to S}$, are then combined to form the final correspondences. From these, $K$ correspondences are selected by thresholding the confidence maps and sampling according to a uniform distribution rule (Edstedt et al., 2024). These correspondences are then used to compute the homography with a robust estimator such as RANSAC (Fischler & Bolles, 1981).

### 3.4 Loss Function

For flow estimation, we apply supervision on the $L_2$ distance between the ground truth grid flow $\mathbf{w}_{gt}$ and the predicted flow $\mathbf{w}$ at every scale $l \in [1, L]$ and iteration $n \in [1, N]$:

$$L_{flow} = \sum_{l=1}^{L} \frac{\mathbf{m}_{gt}^l}{G^l \times G^l} \sum_{n=1}^{N} \lambda^{(N-n)} \rho(\|\mathbf{w}^{l,n} - \mathbf{w}_{gt}^l\|_2), \quad (5)$$

where $G^l$ represents the grid size at scale $l$, and $\lambda \in (0, 1)$ assigns higher weights to later iterations, similar to the "learning to optimize" approach (Teed & Deng, 2020). $\rho$ is a robust cost function that mitigates the impact of outliers (Barron, 2019), as employed in RoMa (Edstedt et al., 2024). $\mathbf{m}_{gt} \in \{0, 1\}$ is a binary mask that indicates which pixels in $I_S$ have correspondences in $I_T$.

For the confidence map, we learn it using a Binary Cross-Entropy (BCE) loss:

$$L_{conf} = \sum_{l=1}^{L} \frac{1}{G^l \times G^l} \sum_{n=1}^{N} \lambda^{(N-n)} \text{BCE}(\mathbf{m}^{l,n}, \mathbf{m}_{gt}^l). \tag{6}$$

Then the total loss is defined as $L = L_{flow} + \alpha L_{conf}$, where $\alpha$ is a hyperparameter balancing the two terms. Notably, all predicted $\mathbf{w}^{l,n}$ are in image space, while the confidence maps $\mathbf{m}^{l,n}$ are in log space, contributing to more stable training.

### 3.5 DATA GENERATION FOR SELF-SUPERVISED LEARNING

**Composite homography**. Following prior works (DeTone et al., 2016; Zhang & Ma, 2024), as shown in Figure 4, we generate synthetic homography datasets for both training and testing. Specifically, we first define a deformation area by selecting four squares at the image's corners, with a deformation ratio $d \in (0, 0.5)$. Here, $d$ is the ratio between the deformation area height and the image height. From each of these squares, four random points are selected and transformed to the center of the respective squares, yielding a homography matrix. This process is performed independently on each image in the pair, producing two homography matrices, $\mathbf{H}_1$ and $\mathbf{H}_2$. The images $I_S$ and $I_T$ are then generated by cropping the

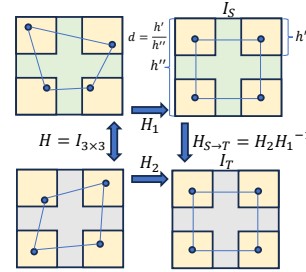

Figure 4: Data generation.

deformed images, with the centers of the deformation areas serving as the corners of the squares. The homography between $I_S$ and $I_T$ is then computed as $\mathbf{H}_{S \to T} = \mathbf{H}_2 \mathbf{H}_1^{-1}$. Different from prior homography estimation approaches which use only a single homography (*i.e.*, setting $\mathbf{H}_2 = \mathbf{I}_{3\times3}$), we introduce two homographies and use their composite to generate training data. This modification is crucial for ensuring that our method remains invariant to the input sequence.

**Augment with dynamic occlusions**. To enable GFNet to handle scenarios with dynamic occlusions, we follow Truong et al. (2023), which augments training data with dynamic objects. Specifically, we treat the images generated in Figure 4 as background. Objects are first added to $I_S$, then transformed with new planar transformations and added to $I_T$ as the foreground, simulating moving occlusions in real-world scenarios. In this case, the ground truth confidence map $\mathbf{m}_{gt}$ is also updated, where the area occupied by the object in $I_S$ and its reprojection from $I_T$ to $I_S$ are set to 0. Moving objects are from MSCOCO (Lin et al., 2014) and augmented training examples are shown in Figure 5.

## 4 EXPERIMENT

### 4.1 DATASETS

**Basic training**. Since the initial grid size is determined by the input resolution, it is preferable to use training data with sufficient resolution to fully leverage our architecture's potential. Following Truong et al. (2020), we utilize the CityScapes (Cordts et al., 2016) and ADE-20K (Zhou et al., 2019) datasets, both of which contain images larger than $750 \times 750$. We generate $33,398$ training samples offline, following the pipeline in Sec. 3.5. The model trained on this split serves as the basic model, which is subsequently fine-tuned for application on multimodal datasets.

**Fine-tuning**. Following Zhao et al. (2021); Cao et al. (2022; 2023); Zhang et al. (2023); Zhang & Ma (2024), we select GoogleMap and VIS-IR as the multimodal scenarios for evaluation. Specifically, GoogleMap consists of paired satellite and map images provided by the Google Static Map API. We create $5,000$ aligned image pairs, each of size $1280 \times 1280$, from various countries for fine-tuning. VIS-IR (Sun et al., 2022) includes $640 \times 512$ RGB and infrared image pairs captured by UAVs, with train, validation, and test splits. We select $5,000$ aligned image pairs from the train spilt for fine-tuning. The final training samples for GoogleMap and VIS-IR fine-tuning are generated on-the-fly from the selected aligned image pairs.

**Inference**. We test on MSCOCO, VIS-IR, GoogleMap and VIRAT datasets, examples are shown in Figure 5. For VIS-IR, we generate test samples from its test split, and for GoogleMap, test samples are generated from regions not overlapping with the training data. VIS-IR and GoogleMap are used to assess the model's ability to handle photometric inconsistencies caused by modality changes, while MSCOCO (Lin et al., 2014) serves as a standard unimodal dataset, commonly used in previous works for evaluation. For each of these three datasets, we generate 1,000 test samples.

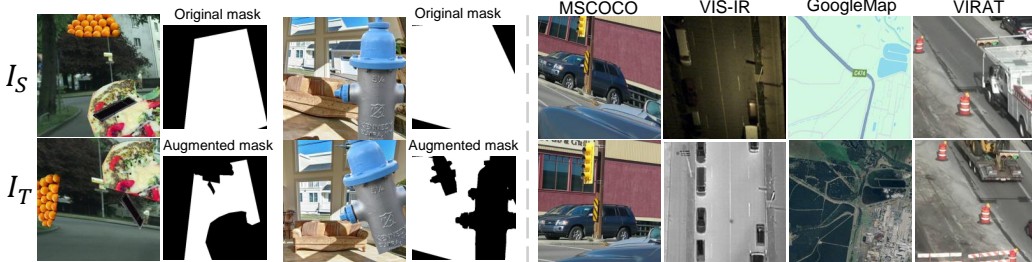

Figure 5: Left: training samples. Right: test samples. In the mask, white for $1$, while black for $0$.

VIRAT is a dataset featuring dynamic scenes captured in a surveillance context (Oh et al., 2011), which we use to evaluate performance on dynamic scenes with occlusions. To create the dataset, we utilize the provided object annotations to identify dynamic objects in the videos. Image pairs are generated by cropping patches centered on the dynamic objects and aligning them with the same patches from different timestamps where the dynamic objects are absent. After removing similar samples, $144$ test samples are finally collected from various video scenarios.

## 4.2 EXPERIMENTAL SETTINGS

**Implementation details**. We use a batch size of $16$ with a learning rate of $2 \cdot 10^{-4}$ for training the basic model, and $1 \cdot 10^{-4}$ for fine-tuning. We use the AdamW optimizer with a weight-decay factor of $10^{-2}$, and use CosineAnnealing to schedule the learning rate. Both the basic training and fine-tuning stages use $2,000,000$ training samples drawn from their respective datasets. Each epoch processes $25,000$ samples, resulting in a total of $80$ epochs. We train at a resolution of $448 \times 448$, with the higher resolution in the recurrence set to $560 \times 560$. We test MSCOCO and VIRAT with the basic model, while VIS-IR and GoogleMap with the fine-tuned models.

For data generation, we set $d = 0.3$, in line with prior works. Composite homography is used in training but not in testing. For the model, DINOv2-large is selected and we stack $4$ cross-attention layers. Decoder at each scale is composed of $8$ stacked depthwise convolution blocks, in line with Edstedt et al. (2023; 2024). The local radius from coarse to fine levels is set to $[7, 6, 4, 2, 0]$. We set $N = 2$ for iterations to manage computational costs, and iteration is performed only on multimodal cases. For the loss, we set $\lambda = 0.85, \alpha = 0.01$. For estimation, we select $K = 5,000$ matches, and use RANSAC with the default setting of cv2.findHomography to calculate homography.

**Metrics**. We calculate the mean reprojection error of the four corner points and report the area under the cumulative curve (AUC) at four different thresholds: 3 px, 5 px, 10 px, and 20 px.

## 4.3 COMPARATIVE RESULTS

**Baselines**. Comparative methods are divided into two groups: homography estimation methods (Group A) and image matching methods (Group B). In Group A, we compare with RHWF (Cao et al., 2023), PRISE (Zhang et al., 2023), and MCNet (Zhu et al., 2024). RHWF and MCNet obtain the homography using a global aggregation layer within the network, while PRISE leverages IC-LK, initialized with MHN (Erlik Nowruzi et al., 2017). In Group B, we compare with the sparse matcher SIFT+LightGlue (Lindenberger et al., 2023), the semi-dense matchers LoFTR (Sun et al., 2021) and Efficient LoFTR (Wang et al., 2024b), and the dense matcher RoMa (Edstedt et al., 2024). For a fair comparison, all methods in Group A are trained under the same settings as ours. For methods in Group B (excluding SIFT+LightGlue), we fine-tune their official outdoor models using our training data. During evaluation, we use RANSAC with the same settings to compute the homography, and the maximum number of correspondences for sparse and semi-dense matchers is set to $2,048$.

**Evaluation on MSCOCO**. As shown in Table 2, GFNet outperforms all comparative methods on MSCOCO. Compared to methods in Group A, GFNet's ability to train at higher resolutions, beyond the $128 \times 128$ limitation of Group A methods, leads to improved accuracy. Compared to methods in Group B, GFNet benefits from the accuracy of the dense matching framework, while the grid-based strategy is highly effective for homography estimation.

Table 2: Comparative results. **Bold**: best, underline: second. Group A: homography estimation. Group B: image matching. Relative improvement or decrease is shown for GFNet.

| Group | Method | MSCOCO | | | | VIRAT | | | |
|---|---|---|---|---|---|---|---|---|---|
| | | auc@3 | auc@5 | auc@10 | auc@20 | auc@3 | auc@5 | auc@10 | auc@20 |
| A | RHWF | 93.02 | 95.81 | 97.9 | 98.77 | 46.57 | 53.7 | 61.53 | 70.64 |
| | MCNet | 91.23 | 94.73 | 97.36 | 98.68 | 42.41 | 50.76 | 59.92 | 69.83 |
| | PRISE | 88.64 | 92.83 | 96.23 | 97.14 | 41.44 | 49.36 | 59.12 | 69.09 |
| | GFNet (ours) | **97.69 (+5.0%)** | **98.61 (+2.9%)** | **99.3 (+1.4%)** | **99.65 (+0.8%)** | **46.92 (+0.7%)** | **54.47(+1.4%)** | **61.73(+0.3%)** | **70.78(+0.1%)** |
| B | SIFT+LightGlue | 89.43 | 93.48 | 96.64 | 98.27 | 38.25 | 47.15 | 57.67 | 68.14 |
| | LoFTR | 94.19 | 96.51 | 98.25 | 99.12 | 42.29 | 50.61 | 59.54 | 69.26 |
| | Efficient LoFTR | 95.65 | 97.39 | 98.69 | 99.34 | 43.34 | 51.17 | 60.13 | 69.83 |
| | RoMa | 96.01 | 97.61 | 98.8 | 99.4 | **47.5** | 54.39 | **61.91** | **70.88** |
| | GFNet (ours) | **97.69 (+1.7%)** | **98.61 (+1.0%)** | **99.3 (+0.5%)** | **99.65 (+0.2%)** | 46.92 (-1.2%) | **54.47(+0.1%)** | 61.73(-0.2%) | 70.78(-0.1%) |

| Group | Method | VIS-IR | | | | GoogleMap | | | |
|---|---|---|---|---|---|---|---|---|---|
| | | auc@3 | auc@5 | auc@10 | auc@20 | auc@3 | auc@5 | auc@10 | auc@20 |
| A | RHWF | 18.16 | 32.05 | 54.48 | 72.84 | 18.32 | 37.47 | 61.09 | 76.43 |
| | MCNet | 16.1 | 30.59 | 51.51 | 71.82 | 16.31 | 35.28 | 59.51 | 75.63 |
| | PRISE | 13.28 | 24.2 | 48.6 | 66.6 | 11.41 | 31.68 | 54.78 | 70.54 |
| | GFNet (ours) | **20.74(+14.2%)** | **33.36(+4.0%)** | **55.1(+1.1%)** | **73.72(+1.2%)** | **49.79(+171.7%)** | **65.67(+75.2%)** | **81.04(+32.6%)** | **89.66(+17.3%)** |
| B | SIFT+LightGlue | 2.56 | 8.91 | 23.33 | 39.99 | - | - | - | - |
| | LoFTR | 10.17 | 22.51 | 45.93 | 65.84 | 4.99 | 13.87 | 32.27 | 51.24 |
| | Efficient LoFTR | 13.28 | 26.32 | 49.43 | 69.1 | 5.37 | 14.49 | 33.6 | 52.73 |
| | RoMa | **21.35** | **34.68** | **55.88** | 73.71 | 45.29 | 62.18 | 78.63 | 88.23 |
| | GFNet (ours) | 20.74(-2.8%) | 33.36(-3.8%) | 55.1(-1.3%) | **73.72(+0.01%)** | **49.79(+9.9%)** | **65.67(+5.6%)** | **81.04(+3.0%)** | **89.66(+1.6%)** |

| | Group A | | | Group B | | | | GFNet (ours) | GFNet (ours) |
|---|---|---|---|---|---|---|---|---|---|
| | RHWF | MCNet | PRISE | SIFT+LightGlue | LoFTR | Efficient LoFTR | RoMa | w/o iteration | w/ iteration |
| Learnable parameters | 1.29M | 0.85M | 19.24M | 11.88M | 11.56M | 16.02M | 111.28M | 3.86M | 3.86M |
| MACs | 23.43G | 4.56G | 55.25G | 38.51G | 249.72G | 179.81G | 2503.19G | 1657.62G | 1709.09G |
| Runtime | 119.70ms | 28.11ms | 204.67ms | 590.11ms | 42.68ms | 31.29ms | 326.57ms | 186.71ms | 209.05ms |

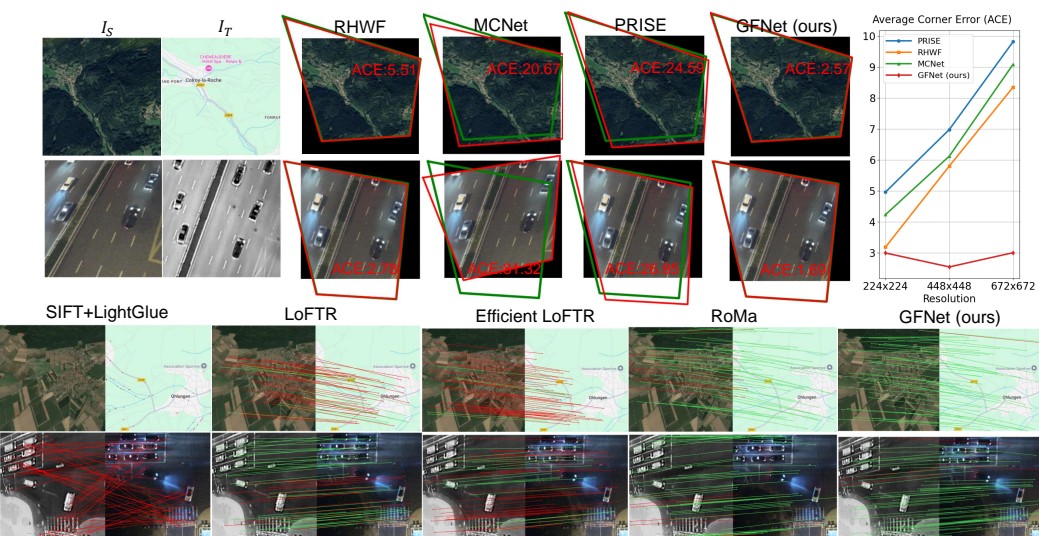

Figure 6: Visualization results on GoogleMap and VIS-IR. Top: homography estimation results. The green polygon is the ground-truth location of $I_S$ on $I_T$, while the red one is the predicted location. The right plot shows ACE at different resolutions on GoogleMap. Bottom: image matching results. We randomly visualize 50 predicted matches. Matches with reprojection error lower than 3 pixels are drawn in green, otherwise red. Only keypoints will be displayed when no matches are predicted.

**Evaluation on multimodal datasets**. GFNet demonstrates significant improvements over SOTA homography estimation methods, with a 14.2% auc@3 increase on VIS-IR and a 1.7× auc@3 improvement on GoogleMap. Beyond the advantage of high-resolution training, this improvement is largely due to the integration of DINOv2, which excels in cross-domain feature matching, boosting performance in multimodal scenarios.

Regarding image matching methods, GFNet surpasses both sparse and semi-dense matchers on the two multimodal datasets. Visualizations are shown in Figure 6. We can see SIFT+LightGlue is unable to handle significant appearance differences between modalities, while LoFTR methods have results with low accuracy. For RoMa, GFNet surpasses it on GoogleMap by 9.9% auc@3 and achieves comparable performance on VIS-IR. The strong improvement on GoogleMap can be attributed to GFNet's iterative strategy, as discussed in the ablation study in Sec. 4.4.

Figure 7: Visualization of inlier information in VIRAT. Darker indicates lower values, and $1 - |I_{T \to S} - I_S|$ is GT. While other methods recognize occlusion implicitly, GFNet handles it explicitly.

Table 3: Ablation on the iterative strategy.

| N | MSCOCO | | | | GoogleMap | | | | MACs |
|---|---|---|---|---|---|---|---|---|---|
| | auc@3 | auc@5 | auc@10 | auc@20 | auc@3 | auc@5 | auc@10 | auc@20 | |
| (1,1,1,1,1) | 97.26 | 98.35 | 99.17 | 99.58 | 43.91 | 60.99 | 77.53 | 87.45 | 1657.62G |
| **(2,2,2,2,2)** | **95.53** | **97.32** | **98.66** | **99.33** | **49.79** | **65.67** | **81.04** | **89.66** | **1709.09G** |
| (4,4,4,4,4) | 95.65 | 97.39 | 98.69 | 99.34 | 50.47 | 66.58 | 81.41 | 89.73 | 1811.27G |

**Evaluation on datasets with dynamic occlusions.** Generally, image matching methods are trained on the 3D datasets to handle two- or multi-view matching, which gives them an inherent ability to address dynamic occlusion. Thus here we mainly discuss how homography estimation methods, which only involve planar information, learn to manage dynamic occlusion. As shown in Figure 7, RHWF and MCNet aggregate matching information to estimate homography, with occluded areas being implicitly filtered during this process. PRISE aligns the feature maps of two images to find the homography, also learning occlusion implicitly. In contrast, GFNet explicitly predicts a mask to identify occlusion, providing more precise occlusion information, which results in better performance on VIRAT compared to methods that implicitly predict occlusion (see Table 2).

**Computational analysis.** From the bottom table in Table 2, we observe that although GFNet is under the framework of dense matching, it reduces the MACs by 33.7% without iteration and by 31.7% with iteration compared to the SOTA dense matcher RoMa. This reduction is attributed to our smaller network architecture and the grid-based strategy.

**Robustness to resolution.** In addition to the $448 \times 448$ GoogleMap test set presented in Table 2, we create two more test sets, each containing 1000 image pairs, with resolutions of $224 \times 224$ and $672 \times 672$. The results are presented in Figure 6. Since homography estimation methods in Group A only accept $128 \times 128$ inputs, errors are calculated at this scale and rescaled to the original resolution, leading to a near-linear increase in error as resolution grows. In contrast, GFNet does not impose input resolution constraints, making it more robust to resolution changes.

## 4.4 ABLATION STUDIES

Ablation experiments are conducted on MSCOCO (natural images) and GoogleMap (multimodal images). We mainly ablate our network architecture (Sec. 3.1), iterative strategy (Sec. 3.3), evaluation strategy (Sec. 3.3), data generation (Sec. 3.5), and training resolution.

**Iteration.** Results in Table 3 demonstrate that iteration is helpful for multimodal data but not for natural images. We attribute this to the challenge of precisely aligning the same physical positions in multimodal image pairs due to photometric inconsistencies, making only approximate alignment possible. The iterative strategy helps learn the process of optimization by mimicking this approximate alignment. However, for natural images where exact alignment is achievable, the iterative approach, which is designed for approximate alignment, can reduce accuracy. When iteration is performed, increasing the number of iterations leads to higher accuracy. Setting $N = 2$ strikes a balance between accuracy and MACs.

**Data generation.** As illustrated in Figure 8, if we only apply the deformation to $I_S$ during training, the network tends to overfit to the input sequence $(I_S, I_T)$ and fails to work on the reversed sequence

Table 4: Ablation on the network architecture and evaluation protocol. When DINOv2 is canceled, we go on one more downsampling in the naive FPN with $128$ channels. "-" means the model fails to converge during training.

| Ablation part | Setting | MSCOCO | | | | | GoogleMap | | | | |
|---|---|---|---|---|---|---|---|---|---|---|---|
| | | auc@3 | auc@5 | auc@10 | auc@20 | MACs | auc@3 | auc@5 | auc@10 | auc@20 | MACs |
| Network | w/o DINOv2 | 95.42 | 97.25 | 98.62 | 99.31 | 63.42G | - | - | - | - | - |
| architecture | w/o Cross-Attention | 97.11 | 98.26 | 99.13 | 99.56 | 1656.58G | 48.18 | 64.48 | 79.66 | 88.61 | 1708.06G |
| Evaluation | w/o symmetric | 97.19 | 98.31 | 99.15 | 99.57 | 1631.03G | 48.80 | 65.08 | 80.56 | 89.34 | 1656.77G |
| protocol | w/o adaptive recurrence | **98.09** | **98.85** | **99.42** | **99.71** | 650.93G | 43.51 | 60.20 | 77.06 | 87.18 | 675.16G |
| | Baseline | 97.33 | 98.40 | 99.20 | 99.60 | 1657.62G | **49.79** | **65.67** | **81.04** | **89.66** | 1709.09G |

Table 5: Ablation on the training resolution. MSCOCO: w/o iteration. GoogleMap: w/ iteration.

| Training resolution | MSCOCO | | | | | GoogleMap | | | | |
|---|---|---|---|---|---|---|---|---|---|---|
| | auc@3 | auc@5 | auc@10 | auc@20 | MACs | auc@3 | auc@5 | auc@10 | auc@20 | MACs |
| 336×336 | 94.26 | 96.38 | 98.86 | 99.01 | 1010.38G | 43.60 | 60.54 | 76.81 | 86.21 | 1049.09G |
| **448×448** | **97.26** | **98.35** | **99.17** | **99.58** | **1657.62G** | **50.55** | **66.48** | **81.29** | **89.73** | **1709.09G** |
| 560×560 | 97.23 | 98.46 | 99.27 | 99.58 | 2022.77G | 50.66 | 65.48 | 79.46 | 87.99 | 2109.12G |

$(I_T, I_S)$. However, by incorporating the composition homography during training, the network learns to generalize to both input sequences.

**Network architecture**. As shown in Table 4, the powerful visual features extracted by DINOv2 lead to an improvement in accuracy on MSCOCO (+2.0% auc@3). For GoogleMap, the training fails to converge when using only the lightweight FPN due to the large modality gap. However, with the incorporation of DINOv2, which provides robust cross-domain visual features, the model is able to successfully converge. Meanwhile, DINOv2 accounts for the majority of the com-

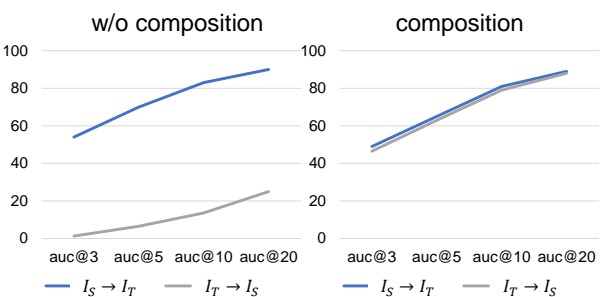

Figure 8: Impact of data generation during training.

putational overhead in our model. Regarding cross attention layers, it provides a stable accuracy improvement on MSCOCO (+0.2% auc@3) and GoogleMap (+3.3% auc@3).

**Evaluation protocol**. As shown in Table 4, symmetry and adaptive recurrence have a limited impact on MSCOCO, as the matches predicted in the initial stage are already sufficiently accurate. However, this approach proves highly effective on GoogleMap, yielding a significant improvement of +2.0% auc@3 with symmetry, and +14.4% auc@3 with adaptive recurrence.

**Training resolutions**. Since the grid size in our approach is determined by the input image resolution, we conduct experiments using different training resolutions to evaluate their impact on the trade-off between accuracy and efficiency. The results, shown in Table 5, indicate that training with a resolution of $448 \times 448$ provides a favorable balance.

## 5  CONCLUSIONS

This paper introduces GFNet, a method designed to deliver high-accuracy homography estimation across varying image resolutions, overcoming the resolution limitations of current approaches. This strong performance stems from the dense matching paradigm and several key components, including the integration of DINOv2, a grid-based strategy, iterative regression, and adaptive resolution recurrence. Experiments show that GFNet outperforms existing homography estimation methods on several challenging datasets, including multimodal and dynamic scenes, and achieves comparable results to the SOTA dense matcher with significantly higher efficiency.

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
