# OpenReview forum: "GFNet: Homography Estimation via Grid Flow Regression"
_ICLR.cc/2025/Conference — ICLR 2025 Conference Withdrawn Submission_

### Official Review · Reviewer_T19m · 2024-10-29

**Soundness:** 3
**Presentation:** 3
**Contribution:** 2
**Rating:** 3
**Confidence:** 5

**Summary:**

This work argues that current methods are constrained to limit resolution which hinders the accuracy of estimating homography, thus proposes a framework to estiamte homography which could utilize the large resolution information. To achieve it, authors firstly compute the flow fileds and then solve homography. As shown in the experiments, GFNet achives multiple sota on common, multimodal, and dynamic datasets.

**Strengths:**

1. The paper is well written and easy to follow.
2. This work is a fairly good system paper.
3. The authors provide the reproduction code and weights in the supplementary material.

**Weaknesses:**

Related Work:

1. Several deep homography methods have been proposed, for example, [1-3]. [1] estimates homography across resolution and modality; [2] generates a supervised homography dataset that simulates the real-world distribution; [3] directly represents the homography via an 8-rank flow field, eliminating the need for post-solving from flow fields.

Method Part:

1. The technical contribution (GRID FLOW REGRESSION) includes initially estimating the flow, then solving for homography, and learning the flow motion between fixed grids. The former seems to has been applied in previous methods [2]. I would like to discuss with the authors the advantages compared to it; the latter, in my view, can be more regarded as a useful engineering trick rather than a major technique contribution.

2. Leveraging priors in foundation models (DINO or Stable Diffusion) to produce features and estimate geometric transformations has also been seen in previous works, such as [4]. These works demonstrate that features from foundation models are very helpful for multi-modality tasks, such as semantic matching.

3. The proposed dataset generating method may not meet the realism criteria (the realism of frame content and inter-frame motion, please refer to [3] for more details), which is crucial for ensuring performance and generalizability. For example, the proposed method cannot simulate parallax changes or human walking.

Experiments:
1. I recommend that the authors conduct experiments, at least zero-shot inference, on recent deep homography datasets [2], which represent general real-world scenes, including parallax changes, dynamic foregrounds, and adverse conditions.

[1] CrossHomo: Cross-Modality and Cross-Resolution Homography Estimation. TPAMI 2024

[2] Supervised Homography Learning with Realistic Dataset Generation. ICCV 2023

[3] Unsupervised Global and Local Homography Estimation with Motion Basis Learning. TPAMI 2023

[4] Emergent Correspondence from Image Diffusion. NIPS 2023

**Questions:**

Motivation:
1. Theoretically, traditional methods can solve a homography with just 4 correspondences. While additional correspondences can enhance accuracy through an overdetermined solution, which is not that essential compared to searching 4 key correspondences.
My question, therefore, is how this approach compares to methods that focus on identifying key correspondences, and why is increasing resolution important, despite the theoretical sufficiency of four correspondences?

---

### Official Review · Reviewer_Jqqw · 2024-10-31

**Soundness:** 2
**Presentation:** 3
**Contribution:** 2
**Rating:** 3
**Confidence:** 5

**Summary:**

This paper proposed a Grid Flow regression Network called GFNet. Compared with the previous methods for homography estimation, GFNet supports varying resolutions and significantly reduces the computational load by using the grid flow regression. When evaluated on several challenging datasets, including multimodal and dynamic scenes, the GFNet achieves state-of-the-art performance with varying resolution and lower computation load.

**Strengths:**

1. The paper provides comprehensive comparisons across various algorithms on multiple datasets, demonstrating that the proposed method achieves superior performance over existing approaches.

2. By incorporating composite homography in data generation, this approach enhances the network's ability to generalize across both forward and reverse input sequences, reducing overfitting and improving robustness under varied conditions.

**Weaknesses:**

1. While the paper mentions that iterative grid flow regression reduces the computation load of global correlation at each scale, the improvement appears limited. The time complexity remains in the same order as pixel-based regression. To clarify the extent of the efficiency gain, it would be useful to see a detailed breakdown of computation time across different components or empirical runtime comparisons on various hardware configurations.

2. I couldn’t find any clear ablation study examining the impact of using grid flow regression on both accuracy and computational load. A comparison between the proposed grid-based approach and a pixel-based version, with other components held constant, could provide valuable insights.

3. While the grid flow regression aims to reduce the computational load, the addition of DINOv2 seems to offset this benefit, potentially increasing the overall computation. To better understand the computational trade-offs, it would be helpful to provide a detailed analysis, including a breakdown of computational costs for each component and how they balance out in the overall architecture.

4. To better clarify the novel contributions of GFNet beyond the use of grid flow and global correlation computation, it would be useful to provide a more detailed comparison between GFNet's iterative structure and that of MCNet.

5. The experimental results are not convincing enough, as many compared methods report their accuracies of MACE (in pixels), such as RHWF, MCNet, and PRISE, but this paper doesn't compare them. The results in Table 2 in this paper aren't consistent with the reported ones in the paper of RHWF, MCNet, and PRISE, as their accuracy is very high on MSCOCO and GoogleMap. Including MACE comparisons and addressing the discrepancies with the results reported in the RHWF, MCNet, and PRISE papers could enhance the clarity and robustness of the evaluation.

**Questions:**

Is there an ablation study that validates the effectiveness of the Augment with dynamic occlusions component?

---

### Official Review · Reviewer_SDxP · 2024-10-31

**Soundness:** 3
**Presentation:** 3
**Contribution:** 2
**Rating:** 5
**Confidence:** 5

**Summary:**

This work proposes a grid flow regression network that consistently delivers high-accuracy homography estimates across varying image resolutions. In particular, the proposed GFNet directly predicts the flow over a coarse grid and then uses the resulting correspondences to obtain the homography. Experimental results demonstrate the effectiveness of the proposed method across various datasets.

**Strengths:**

- This work serves as the first attempt to address the limitation of the fixed input resolution in the homography estimation problem. The motivation of using a grid flow-like representation is clear and sound.
- The paper is easy to read and the structure is well organized.
- The experiments are comprehensive and the results are convincing. The proposed method outperforms previous methods across various datatasets, showing promising performance and robustness.
- The codes and pre-trained models have been uploaded, which significantly helps the reviewers to understand the details and mechanism of the proposed framework.

**Weaknesses:**

However, I still have some concerns as follows and tend to raise my rating if the authors can properly address them.
- The proposed grid flow representation is similar to the classical 'mesh flow' (MeshFlow: Minimum Latency Online Video Stabilization) used in the video stabilization task. Their differences and limitations are expected to be clarified in the context of the homography estimation.
- The pixel-wise flow can flexibly adapt to different input resolutions, but it might be redundant to describe the limited DoF homography matrix, which typically only has 8 DoF. Do we really need such a dense flow in a homography estimation network? More discussions should be presented when applying the flow and grid representations.
- This work introduces an iterative flow regression approach to prevent suboptimal multi-scale flow optimization in challenging scenes. However, this iterative or progressive regression method has also been explored in previous flow estimation and image warping works. For example, "Raft: Recurrent all-pairs field transforms for optical flow", "MOWA: Multiple-in-One Image Warping Model", "Semi-supervised coupled thin-plate spline model for rotation correction and beyond", etc. The authors are suggested to highlight their unique contributions and provide some discussions compared with the above works.
- Some recent homography estimation works are also missing in this work. For example, "DMHomo: Learning Homography with Diffusion Models", "Supervised Homography Learning with Realistic Dataset Generation", "Depth-aware multi-grid deep homography estimation with contextual correlation", etc.
- Is the sota performance of this work mainly derived from the powerful DINOv2 backbone? Did the author try other backbone networks such as the classical ResNet?
- When performing the ablation study, an important experiment would be using the grid or 4-pt representation (widely used in previous homography estimation works) to compare the proposed grid flow representation in different resolution settings, including their specific estimation accuracy, complexity, and efficiency.

**Questions:**

Can the proposed method be applied to the related downstream vision tasks? Some discussions and future works could be provided.

---

### Note · Authors · 2024-11-14

I have read and agree with the venue's withdrawal policy on behalf of myself and my co-authors.